# Computing water flow through complex landscapes, Part 2: Finding hierarchies in depressions and morphological segmentations

Richard Barnes[1,2,3], Kerry L. Callaghan[4], and Andrew D. Wickert[4,5]

[1]Energy & Resources Group (ERG), University of California, Berkeley, USA
[2]Electrical Engineering & Computer Science, University of California, Berkeley, USA
[3]Berkeley Institute for Data Science (BIDS), University of California, Berkeley, USA
[4]Department of Earth & Environmental Sciences, University of Minnesota, Minneapolis, USA
[5]Saint Anthony Falls Laboratory, University of Minnesota, Minneapolis, USA

**Correspondence:** Richard Barnes (richard.barnes@berkeley.edu)

**Abstract.** Depressions—inwardly-draining regions of digital elevation models—present difficulties for terrain analysis and hydrological modeling. Analogous "depressions" also arise in image processing and morphological segmentation, where they may represent noise, features of interest, or both. Here we provide a new data structure—the depression hierarchy—that captures the full topologic and topographic complexity of depressions in a region. We treat depressions as networks in a way that is analogous to surface-water flow paths, in which individual sub-depressions merge together to form meta-depressions in a process that continues until they begin to drain externally. This hierarchy can be used to selectively fill or breach depressions, or to accelerate dynamic models of hydrological flow. Complete, well-commented, open-source code and correctness tests are available on Github and Zenodo.

## 1 Introduction

Depressions (see Lindsay, 2015, for a typology) are inward-draining regions of a DEM that lack an outlet to an ocean, map edge, or some other designated boundary. Quantifying and understanding these depressions and their structure can advance our understanding of wetlands (Wu and Lane, 2016), subglacial hydrology (Humbert et al., 2018) and its links to sea-level rise (Calov et al., 2018), microscale water retention in soils (Valtera and Schaetzl, 2017), and flood extent (Nobre et al., 2016). This is particularly significant because lakes and wetlands host biodiversity, provide ecosystem services including denitrification (Hansen et al., 2018) and recreation (Costanza et al., 2006; Keeler et al., 2015), and impact sediment dynamics (Wickert et al., 2019; Mishra et al., 2019) as well as drainage-network integration (Lai and Anders, 2018; Hilgendorf et al., 2020), and realignment (Carson et al., 2018).

Likewise, in image processing and segmentation, regions of differing image intensity and color can be modeled as depressions that represent either noise or features of interest. In this context, geomorphological algorithms for depression-handling (e.g., Barnes et al., 2014b) have been applied to the cosmic microwave background radiation (Giri et al., 2017), nanoparticle chemistry (Svoboda et al., 2018), biological membranes (Kulbacki et al., 2017), road-car segmentation (Beucher, 1994), murder and crime statistics (Khisha et al., 2017), remote sensing of buildings (Golovanov et al., 2018), neuron map-

ping (Iascone et al., 2018), and metal-defect mapping (Blikhars'kyi and Obukh, 2018). This multidisciplinary set of uses demonstrates the broad potential of a generalized algorithm that can compute depressions and their topology.

Depressions complicate algorithms for geomorphological and terrain analysis, as well as hydrological modeling. Many common methods route flow using only information about local gradients, and enforce downgradient flow (O'Callaghan and Mark, 1984; Mark, 1987; Freeman, 1991; Quinn et al., 1991; Holmgren, 1994; Tarboton, 1997; Seibert and McGlynn, 2007; Orlandini and Moretti, 2009; Peckham, 2013). As a result, flow entering a depression cannot leave; in an extreme case, this could cause a continent-scale river, such as the entire Mississippi, to disappear into a small depression.

Because correctly routing flow in depressions and flat areas requires non-local information, addressing the influence of depressions on hydrological networks requires more work than a simple downslope-routing algorithm. Depressions—especially those in high-resolution datasets—are often treated as aberrations. Algorithms to remove these features either flood them until they are filled and flow paths can reconnect (Barnes et al., 2014b); carve deep channels through them either by modifying the DEM's data directly or by altering flow directions to simulate carving (Lindsay, 2015; Martz and Garbrecht, 1998), as 35 in `r.watershed`; or perform some combination of these two options (Grimaldi et al., 2007; Lindsay and Creed, 2005a; Lindsay, 2015; Schwanghart and Scherler, 2017). However, depressions may also represent actual landscape features such as prairie potholes, lakes, wetlands, and soil microrelief (Shaw et al., 2012, 2013; Valtera and Schaetzl, 2017). When this is the case, depressions should be retained and leveraged to improve models (Callaghan and Wickert, 2019; Arnold, 2010; Hansen et al., 2018).

Incorporating depressions into drainage analyses is non-trivial. Depressions may have complex topographic structure. For instance, Vulcan Point is an island within Main Crater Lake, which is on Taal Island in Lake Taal, which itself is on the island of Luzon in the Philippines. As another example, Lake Nipigon (Ontario, Canada) contains Kelvin Island, which in turn contains Firth Lake, which hosts its own islands. High-resolution data can exacerbate the issue by introducing high-frequency noise that cannot be reliably distinguished from actual topographic features (Lindsay and Creed, 2005b, c).

This problem is similar to one in image processing, in which a computer must reassemble multiple distinct-looking features into a meaningful whole. For example, over-segmentation can cause features such as cars to be fragmented into many small pieces (Beucher, 1994). Understanding the relationships between topographic depressions can aid the general goal of building relational hierarchies among adjacent objects, and in so doing can reduce over-segmentation by providing a principled way of merging small features and extracting composite features of interest.

In response to these challenges, we present an efficient method for constructing a *Depression Hierarchy*: a data structure that captures the full topologic and topographic complexity of depressions in a region. The hierarchy can be used to selectively fill or breach depressions, or to accelerate dynamic models of hydrological flow.

Prior researchers have developed structures with similar purpose—and in some cases, function—to depression hierarchies, but these either yield nondeterministic results, are not developed in a way to permit dynamic water flow through a set of 55 nested depressions, or are prohibitively slow. Beucher (1994) presents a hierarchical segmentation algorithm for images using a "waterfall" approach that merges adjacent features by filling smaller local minima while maintaining significant minima that can act as a sink over larger regions. However, this "waterfall" algorithm does not produce a persistent data structure to be

used in subsequent operations nor does it construct a full hierarchy as an intermediate product. Salembier and Pardas (1994) use a kind of hierarchical segmentation, but generate the hierarchy via repeated simplification of the source image. These simplifications are sufficient to segment features, but, in a hydrological context, can lead to unacceptable degradation of terrain information. Arnold (2010) presents a similar algorithm to the one developed here. However, no source code is provided, the generated hierarchy is not formalized, and the algorithm generates circular topologies that require correction. Wu et al. (2015) and Wu and Lane (2016) develop a method for extracting depression hierarchies by first smoothing a DEM and then extracting vector contour lines from it. They then analyze the topological relationship of the contours. Wu et al. (2018) build on this approach by developing a method to move a horizontal plane upwards through topography and noting the elevations at which depressions combine. Both methods are inaccurate due to their reliance on discrete vertical steps—that is, both the contour intervals and the finite distance over which the plane is shifted upwards before checking for joined depressions. The latter method is also inefficient because it requires every cell of the terrain model to be parsed after each movement of the plane. Cordonnier et al. (2018) present an algorithm based on minimum spanning trees in a planar graph, which can be used to construct a hierarchy of depressions. However, the resulting data structure is not well-described, and the algorithm has been optimized for use in contexts in which the dynamic flow of water (described at greater length in §6.5) does not need to be modeled explicitly. Callaghan and Wickert (2019), in a companion paper to this, describe an approach to move water among cells across the landscape. This virtual water floods depressions, but its cell-by-cell computation is expensive and slow, and the algorithm does does not obtain information on the topological structure of the surface.

The depression hierarchy presented in this paper is differentiated by several features. (1) Correctness: the DEM does not require preprocessing and no arbitrary step length needs to be defined. (2) Efficiency: the algorithm operates in $O(N)$ time. (3) Degree of documentation: in addition to this paper, 48% of the lines in the accompanying source code are or contain comments. (4) Availability of source code: the completed, well-commented source code for the algorithms described here, along with associated makefiles and correctness tests, is available on both Github and Zenodo (Barnes and Callaghan, 2019). (5) Suitability for dynamic models: by defining hydrological connectivity across a landscape, the depression hierarchy can be leveraged to accelerate hydrological models.

## 2  The Depression Hierarchy

The depression hierarchy consists of a forest of binary trees, as shown in Figure 1 and illustrated in Figures 2 and 3. The leaves of the trees are the smallest, most deeply-nested depressions (Figure 2). During flooding, these would fill first. Non-leaf nodes are formed when two depressions overflow into each other. Here, this non-leaf node is termed a "parent" and each of the overflowing depressions—be they leaves or no—is termed a "child". Eventually, a depression fills to the level at which additional "water" would escape the initial set of depressions and flow into either the ocean or another binary tree of depressions that already has a path to the ocean. For example, in Figure 1, node 12 flows into leaf-node 4, which (indirectly) flows into the ocean. When this happens, one binary tree cannot become the child of the other, since they are not topographically nested. Instead, the root (the topmost node) of the tree that does not yet link to the ocean takes one of the leaf nodes of the

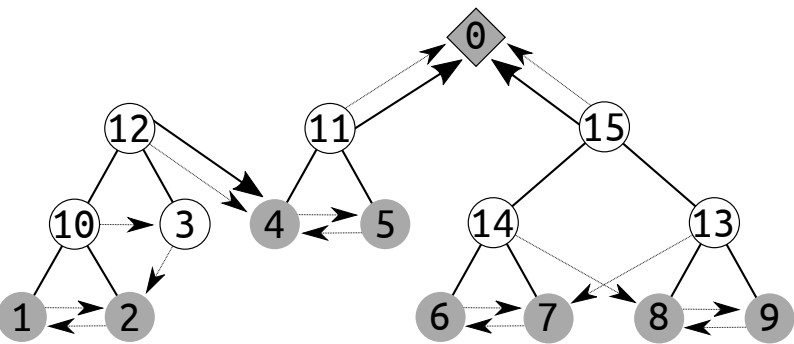

**Figure 1. A depression hierarchy** of the topography depicted in Figure 2 generated by a process shown in Figure 7. Dotted arrows indicate *geolinks*, solid lines indicate links between depressions and meta-depressions, solid arrows indicate *oceanlinks*. (11), (12), and (15) are all *roots* of binary trees. In each of several binary trees, water fills the tree from bottom to top before overflowing into a neighboring tree or the ocean. As (1) fills up, it overflows through its *geolink* (the dotted arrow) into (2). Both of these then begin to fill (10), a larger depression containing both, as indicated by the solid lines between (10) and both (1) and (2). When (10) overflows, it begins to fill (3) through the dotted geolink arrow. When (3) overflows, it tries to fill (2), but finds it full. Therefore, both (3) and (10) begin to fill (12). Topologically, (12) flows into (4); however, the reverse is not true. This is because the depression tree rooted at (12) must actually be uphill of (4). Thus, (12) notes that (4) is its parent (solid arrow) and the depression into which it overflows (geolink, dotted arrow), and (4) makes an *oceanlink* to (12), as implied by the solid arrow, but does not count it as a child. Both (11) and (15) flow into the ocean (0), which may have an infinite number of children. A cross-sectional view of the landscape described by this depression hierarchy is shown in Figure 2.

other tree as its parent and that leaf node makes an *oceanlink* in the reverse direction. In addition to the primary structure of the depression hierarchy (solid lines in Figure 1), we define a set of *geolinks* that tie an overflowing depression to the geophysically neighboring depression into which its overflow ultimately flows. As in a threaded binary tree (Fenner and Loizou, 1984), these links can be used to accelerate traversals by eliminating recursion.

## 3 The Algorithm

The depression-hierarchy algorithm proceeds in several stages, as detailed below: (1) ocean identification, (2) pit-cell identification, (3) depression assignment, and (4) hierarchy construction. As a side effect, the algorithm determines flow directions. Flowcharts showing the steps taken by the algorithm can be found in Figures 4, 5, and 6. We describe the algorithm with reference to Figure 7.

Several bookkeeping data structures are required to compute the depression hierarchy. These are:

– *DEM*: A 2D array indicating the elevation of each cell, or, in the case of image segmentation, its intensity. The data type is arbitrary.

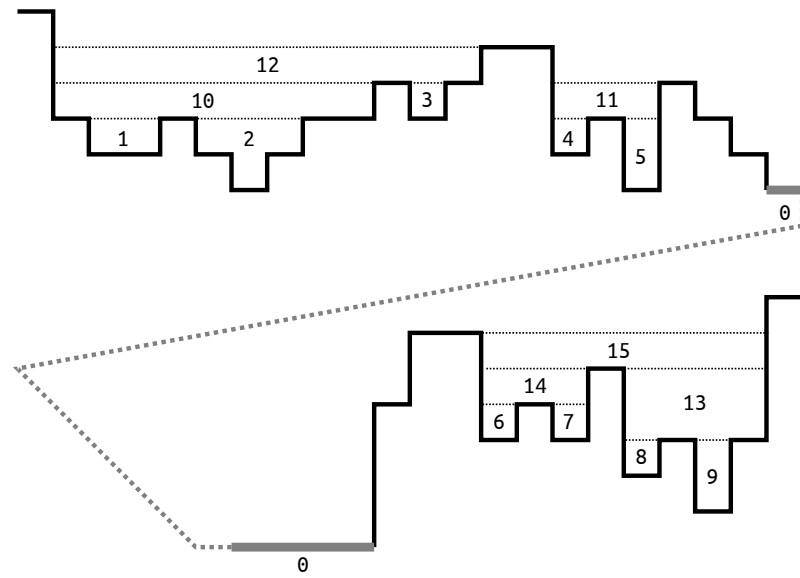

**Figure 2. 1D topography representing the depression hierarchy presented in Figure 1.** Solid black lines represent topography. The thick gray line represents the ocean, and the dotted line indicates that this figure represents a single continuous profile that has been split to better fit on the page. Following Figure 1, numbers mark depressions and meta-depressions, and "0" marks the ocean. A traditional depression-filling flood-fill algorithm would simply fill each depression up to the level of the highest dotted line, thus losing information on the structure within each depression.

– *Label*: An array with the same shape as *DEM* indicating which leaf depression each cell belongs to. Initially, all cells are labeled with the special value NODEP.

– *Flowdir*: An array with the same shape as *DEM* that indicates the flow direction of each cell. Initially, all cells are labeled with the special value NOFLOW. The algorithm returns flow directions as an output. They are determined in a standard way by requiring that each cell direct its flow in a D8 fashion to the lowest of its eight neighbors. In the case that the lowest neighbor is not unique, one is chosen arbitrarily.

– *PQ*: A priority queue that orders cells such that the cell of lowest elevation is always popped (i.e., removed from the queue) first. In the event that two cells have the same elevation, the cell added most recently is popped first.

– *DH*: The depression hierarchy, a forest of binary trees that store the hierarchical relationships among depressions alongside metadata about each depression.

– *OC*: A hash map of depression outlets. The hash map is a relational data structure that links keys to values (Cormen et al., pp. 253–285). Outlets are identified by the two depressions that they join. Therefore, the depressions' identifiers (ids) are used as the hash map's keys, while the associated values contain information such as the elevation of the spillway

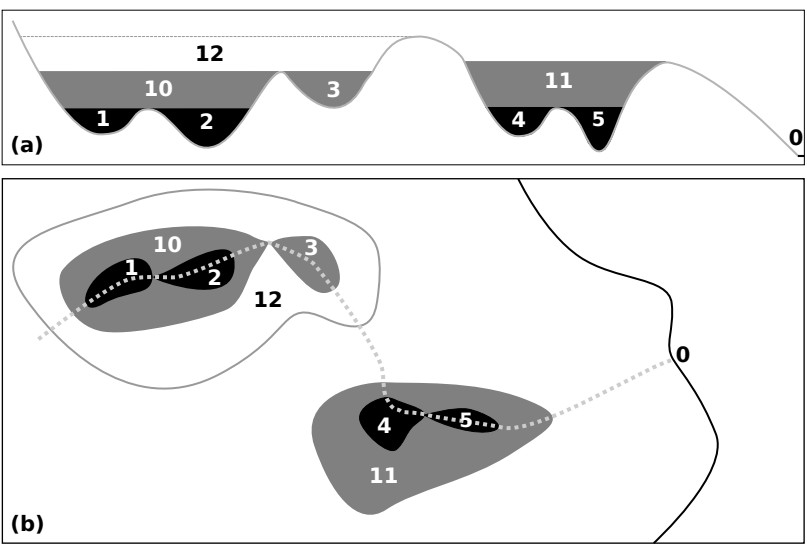

**Figure 3. Cartoon of the left-hand side of the depression hierarchy from Figures 1 and 2.** In this case, we consider depressions to be represented in topography. **(a)** Cross-sectional view. **(b)** Map view. Numbering follows Figures 1 and 2. Flood-fill algorithms used for depression-filling would simply fill these depressions to the highest levels (labelled 11 and 12); the direction of flow and the structure within each depression would not be considered.

sill separating two depressions. Though many potential outlets between two depressions may be found, lower outlets overwrite higher ones such that only the lowest is retained.

– *DS*: A disjoint set data structure (also known as a "union find", "set union", or "merge–find") (Cormen et al., pp. 561–585) is used to quickly determine the root of a tree of depressions.

## 3.1 Ocean Identification

All cells must have a drainage path to the "ocean". This path may be simple and direct when flow down a river terminates directly in an ocean. It can also be indirect, when flow enters a depression, fills the depression, and then spills out towards the ocean, possibly entering more depressions on its way.

The *Label* of all cells that constitute the ocean have the special value OCEAN. For some applications, OCEAN cells can be determined by comparing cells' elevations with a value for sea level (Mitrovica and Milne, 2003, p. 257). In other applications, especially in landlocked regions and image segmentation applications, the edge cells of the DEM can be marked as OCEAN to ensure that flow reaches the edge of the area of interest. A hybrid of these may also be used, in which all cells in contiguous regions of below-sea-level cells that touch an edge of the grid are labeled as ocean; this ensures that below-sea-level basins in the continental interior remain distinct from the ocean (Wickert, 2016).

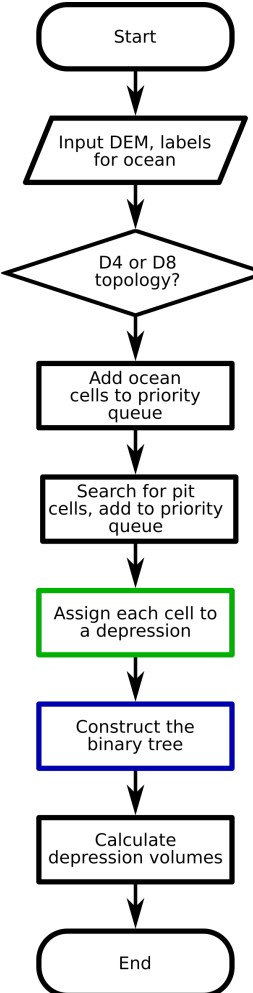

**Figure 4. Main steps taken by the depression hierarchy algorithm.** More detail on the green and blue boxes can be seen in Figure 5 and Figure 6.

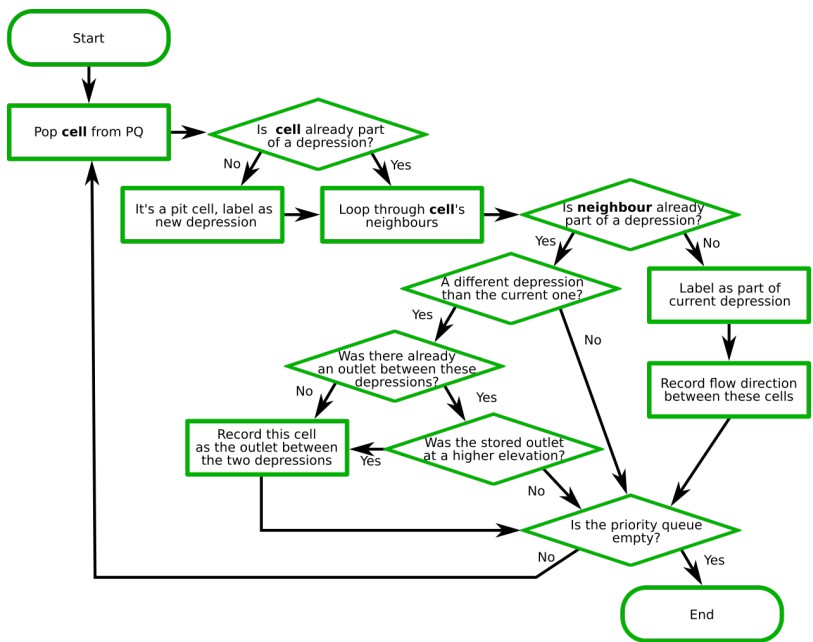

**Figure 5. Cell assignment to unique depressions.** Each cell receives a number that is its *leaf label*. See Figure 4 to find how this process fits into the overall depression-hierarchy algorithm.

130    All ocean cells are added to the priority queue *PQ* as they are identified. A single depression representing the entire ocean is added to *DH*. Figure 7a depicts this initial state before the start of the "flooding" process.

### 3.2 Pit Cell Identification

After the ocean—the ultimate sink—is selected, the depression-hierarchy algorithm must identify all of the pits in the *DEM* that can act as local sinks for water. For the purposes of this paper, a pit cell is a cell that does not drain to any of its neighbours: all of the neighbours' elevations are equal to or greater than that of the pit. All pit cells are added to *PQ* as they are identified, as depicted in Figure 7a.

### 3.3 Depression Assignment

Once all pit and ocean cells are identified, the depression-hierarchy algorithm places them in *PQ*. The general strategy now is to pop (i.e., select and remove) cells from *PQ*, label the popped cells' unlabeled neighbours, add the previously unlabeled neighbours to *PQ*, and repeat this process until *PQ* is empty. Once *PQ* is empty, all of the cells of *DEM* will have been visited. See Figure 5 for a flowchart describing this section of the algorithm. This operation is similar to the Priority-Flood algorithm (Barnes et al., 2014b).

For each cell $c$ that is popped, one of three possibilities must be true:

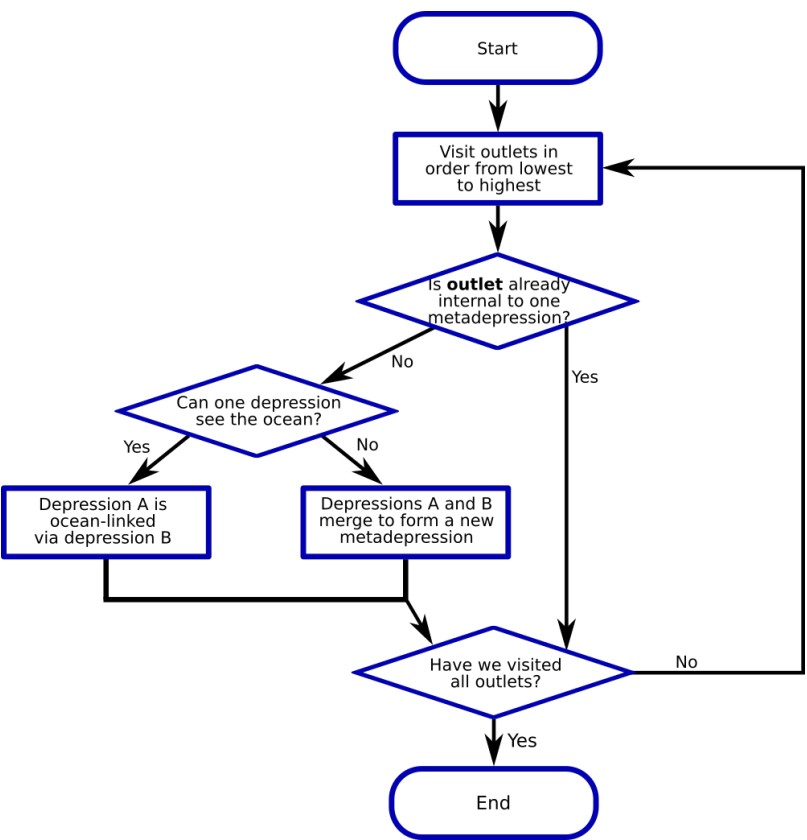

**Figure 6. Construction of the forest of binary trees.** This forest represents the links between individual depressions (Figure 1). See Figure 4 to see how this process fits into the overall depression-hierarchy algorithm.

1. *Label(c)*=OCEAN.

145    2. *Label(c)*=NODEP.

3. Neither of the above.

If *Label(c)*=OCEAN, the cell $c$ is either part of the ocean or has already been proven to flow to the ocean. In this case, nothing more need be done.

If *Label(c)*=NODEP, cell $c$ is a pit cell. Although all cells begin with the NODEP label, a popped cell will label its NODEP
150    neighbours. Therefore, if we pop a NODEP cell, we know that cell does not flow into any existing depression and is the pit of a new one. This is also true of flat areas: *PQ* is designed such that if two cells have the same elevation the cell added most recently is popped first. Therefore, the first cell found in a flat greedily labels every other cell. As each pit cell is found, a new depression is added to *DH* and its label is applied to *Label(c)*. Therefore, for a given map, the choice of which cell becomes

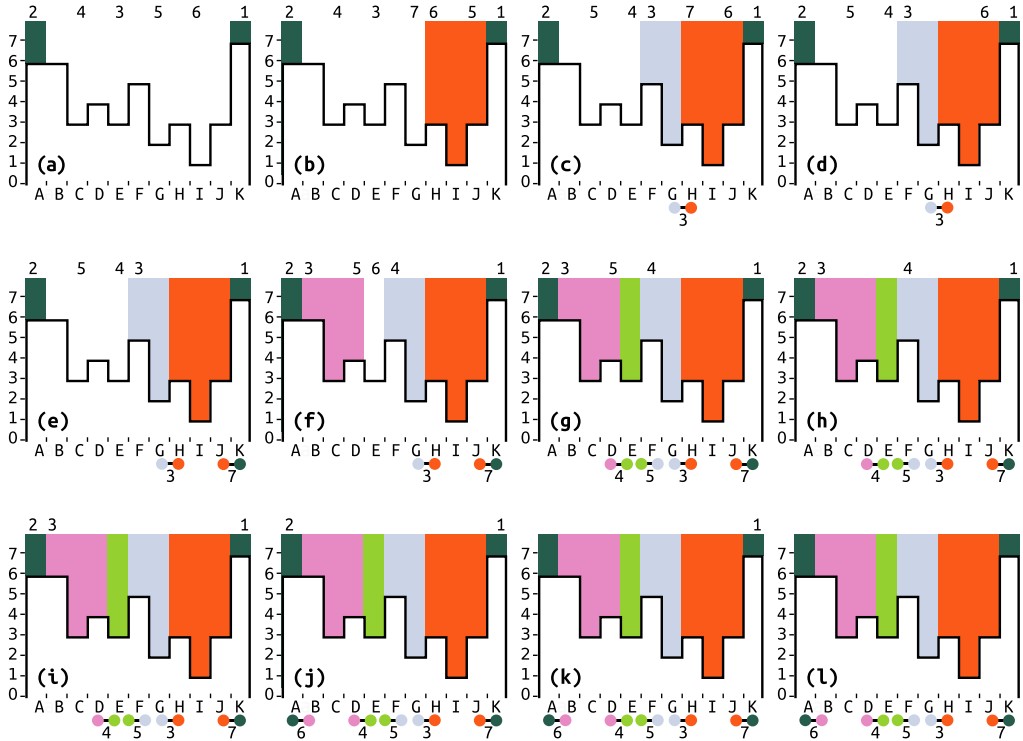

**Figure 7. Illustration of the "Flooding" Process** as applied to the right-hand side of the topography shown in Figure 2. Boldface lowercase letters indicate progression through time. Capital letters label cells. Numbers at the top indicate the cells' positions (if any) in a priority-queue $PQ$. The little barbells indicate outlets between depressions with numbers to indicate their elevations. The black lines outlining the white regions indicate elevation, with values shown on the y-axis. Colors represent labels. **(a)** Initialization. $C$, $E$, $G$, and $I$ are pit cells (they have no lower neighbours), so they are added to PQ. $A$ and $K$ are ocean cells, so they are labeled as such and added to PQ. $I$ is the lowest cell and so has the highest priority. **(b)** $I$ is popped. It is not already labeled, so it is a new depression and given a new label. $H$ and $J$ are labeled and added to PQ. $H$ and $J$ have the same elevation as $C$ and $E$, but since they have been added to the PQ more recently, their priority is higher. Arbitrarily, $H$ is given the higher priority. **(c)** $G$ is popped and given a new label. $H$ shares $I$'s label, so it is ignored. $F$ is labeled and added to PQ. An outlet between $G$ and $H$ is recorded with elevation 3. **(d)** $H$ is popped. It is already labeled, so it is not altered. $H$'s neighbours have already been labeled and so nothing is done to them either. Nothing new is added to the PQ. The outlet between blue and orange has already been noted, so no new outlet is recorded. **(e)** $J$ is popped. It is already labeled, so it is not altered. Its neighbour, $K$, has a different label (ocean), so an outlet of elevation 7 between the two depressions is noted. **(f)** $C$ is popped and given a new label. $B$ and $D$ are a part of $C$'s depression, so they are given its label and added to the PQ. **(g)** $E$ is popped and given a new label. It was not yet labelled, so it is given a new label. Its neighbours were already labeled, so an outlet of elevation 4 is noted between $D$ and $E$, and an outlet of elevation 5 is noted between $E$ and $F$. Nothing new is added to the PQ. **(h)** $D$ is popped. It is already labeled, as are both neighbours. An outlet between pink and green is already recorded, so no new outlet is noted. **(i)** $F$ is popped. It is already labeled, as are both neighbours. An outlet between blue and green is already recorded. **(j)** $B$ is popped. It is already labeled, so it is not relabeled. An outlet to the ocean at elevation 6 is noted. . **(k)** $A$ is popped. It is already labeled, and the outlet between ocean and pink has already been recorded, so nothing happens. **(l)** $K$ is popped. It is already labelled, and the outlet between ocean and orange has already been recorded, so nothing happens. No cells are left in PQ, so the algorithm completes.

the labeled "pit" in a flat area is deterministic based on the regional topographic structure, but the exact cell to have this label is not physically meaningful. Figure 7b, c, f, and g depict situations where new labels are given to cells.

If *Label(c)* is neither OCEAN nor NODEP, cell $c$ has already been assigned to a depression. This means either that: (i) $c$ is on the frontier of the traversal, and will therefore have neighbours that have not yet been seen and must be added to *PQ*, (ii) that $c$ was part of a flat that has already been processed and therefore all its neighbours have been seen and none should be added to *PQ*, or (iii) $c$ is at the edge of a depression and its neighbour has been labelled as a different depression. In this last case, $c$ may be the outlet between the two depressions, if it is the lowest link between them. Figure 7d, e, and h–l represent the third case, in which a previously labeled cell sees neighbours which are part of a different depression. Of these, subfigures e and j include the discovery of a new outlet. We discuss this further below.

After identifying the state of cell $c$ and modifying it as indicated above, *Label(c)* must be either OCEAN or the label of a depression. If it is a depression, it is one of the leaves in the depression hierarchy (gray circles in Figure 1). If it is ocean, we know that it sits at the upper-most end of the depression hierarchy (gray diamond with black border in Figure 1).

From this point, the next step is to consider how the popped cell $c$ interacts with each of its neighbours, $n$. As before, there are three distinct possibilities:

1. *Label(n)*=NODEP.

2. *Label(n)*=*Label(c)*.

3. Neither of the above.

If *Label(n)*=NODEP, $n$ has not previously been seen. Accordingly, *Label(n)* is set to *Label(c)*, $n$ is placed into *PQ*, and *Flowdir(n)* is set to point to $c$. This ensures that flow follows the path of steepest descent since $c$ is the lowest unexplored cell in the DEM. Figure 7b depicts one example of this, in which the previously-unlabeled cells "H" and "J" are labeled as part of the orange depression. Another example, provided in Figure 7c, depicts the previously-unlabeled cell "F" being labeled as a part of the light blue depression.

If *Label(n)*=*Label(c)*, $n$ is skipped because it has either already been visited or has already been added by another cell. This also ensures that flats are processed only once. For example, in Figure 7d, neighbour cell "I" already has the same label as "H", the cell currently being considered, and so "I" is skipped.

If neither of the above is true, *Label(n)*≠NODEP and *Label(n)*≠*Label(c)*. The remaining possibility is that *Label(n)* equals the label of a depression that is distinct from that of its newly-popped neighboring cell, $c$. Therefore, this indicates that two different depressions are meeting. For example, in Figure 7d, neighbour cell "G" already has a different label than "H", the cell currently being considered; therefore, "G" retains its label.

In this final case, we note where two different depressions meet by creating a link between them. To do so, we determine whether the elevation of $n$ or $c$ is higher. The higher of the two is the outlet cell, and its elevation is the depression's spill elevation (that is, the elevation to which water must rise in order to flow out of the depression). The depression-hierarchy algorithm then adds an object containing this information to the hash map *OC*. The contents of *OC* are hashed using the labels

of the depressions that are joined by an outlet. If any entry for an outlet is already present, only the outlet of lower elevation is retained. Two depressions may share a border across multiple cells (i.e. there are multiple potential spillways), but only the location of the lowest outlet is recorded since this is the only location where overflow from one depression to the other would naturally occur. Below, we will transform this set of lowest inter-depression links into geolinks since the set identifies which leaf depressions are geophysical neighbors. Figure 7c, e, g, and j are examples of this, but the one-dimensional elevation profile in Figure 7 cannot depict the case of multiple outlets of different elevation.

After completing this process, the depression assignment algorithm then selects the next cell $c$ from the priority queue and repeats the above set of steps until $PQ$ contains no more cells. Upon completion of the depression assignment phase, the algorithm will have visited and labeled all of the cells, assigned each of them a flow direction, and identified the lowest outlet between each adjacent pair of depressions. See Figures 9 and 10 for examples of how the current labels would appear.

## 3.4 Hierarchy Construction

At this point *Label* associates every cell with the label of a depression corresponding to an entry in *DH*. These entries will form the leaves of the depression hierarchy (gray circles in Figure 1). Each depression contains all of the cells lower than its spill elevation as well as all cells whose flow ultimately terminates somewhere within the depression. Such a set of cells can also be termed a "basin" (Cordonnier et al., 2018). Figure 8a depicts this.

The next order of business is to identify the structure of flow among the depressions. Pairs of depressions that flow into one another—that is, those connected by links in Figure 7—will join to form meta-depressions. The elevations of these meta-depressions extend from the spill elevation (i.e. the height of the sill) between the two depressions to the elevation of the next-highest contiguous sill. Pairs of leaf depressions, meta-depressions, or a leaf and a meta-depression can join to form higher-order meta-depressions recursively to represent the structure of depressions in the landscape.

Not all depressions flow into each other because the binary tree stops growing when its root finds an outlet to the ocean. Therefore, *DH* is a forest of binary trees, where "forest" refers to the fact that multiple binary trees of depressions and meta-depressions may exist that do not link directly. See Figure 6 for a flowchart describing this section of the algorithm.

All outlets are labeled with reference to the leaves of the binary trees. However, some outlets will drain meta-depressions rather than the leaf depressions that have been used to label the outlets. As an example, in Figure 8, 5 drains into 6, but the cells that actually constitute the outlet will be labeled 2 and 3.

A fast way to determine the hierarchical structure of a depression set—such as determining that depression 6 in Figure 8 contains depression 2—is to implement a disjoint-set data structure (Galler and Fischer, 1964; Tarjan and van Leeuwen, 1984). A disjoint set, also known as a "union find", "set union", or "merge-find", quickly identifies which of its elements belong to the same set. In the case of the depression hierarchy, each depression is an element of the disjoint set, and each of these elements is initially marked as being its own set. Pairs of these sets may be merged such that one set becomes the parent of another. Repeating these merges forms the aforementioned forest of trees.

Merges in a disjoint set are usually performed using "union by rank", but this discards information that is critical to building a depression hierarchy. When combining depressions following "union by rank", the shorter tree is made a child of the taller

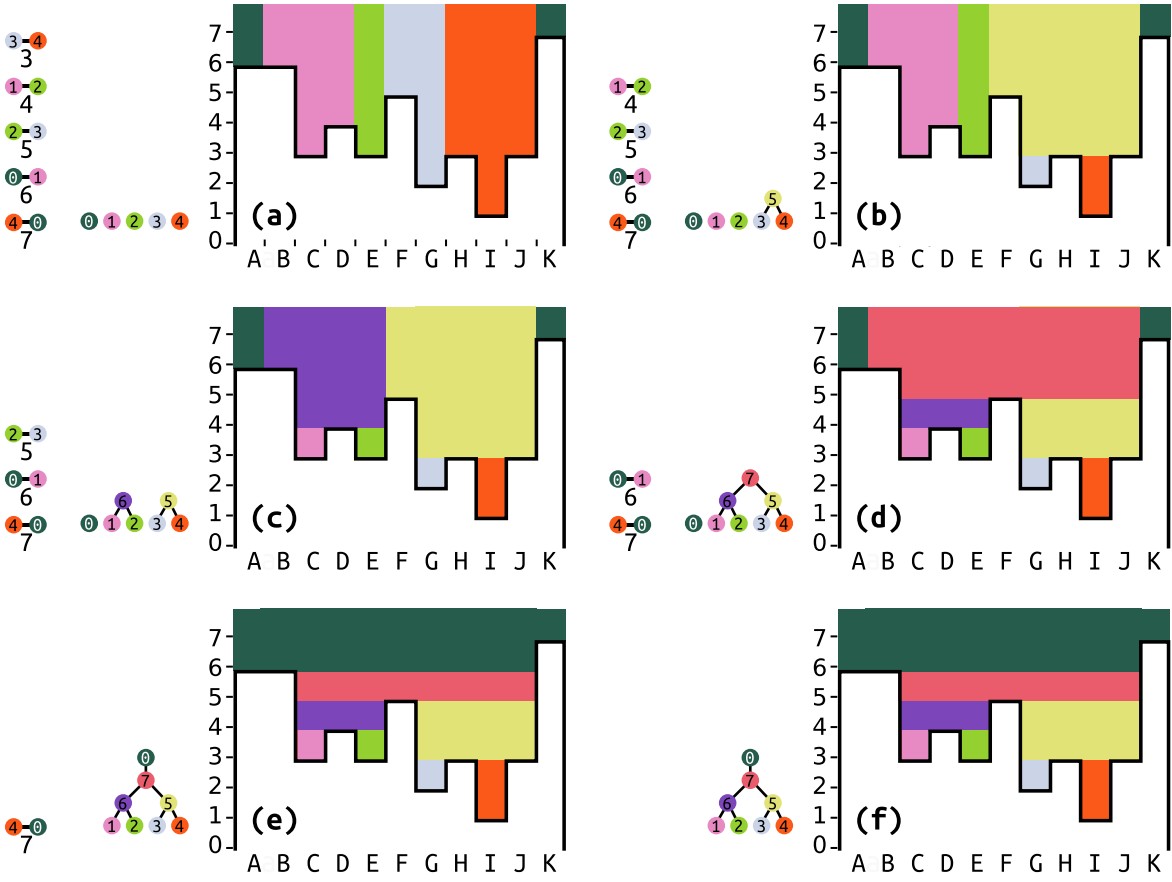

**Figure 8. Illustration of the Hierarchy Construction.** Boldface lowercase letters indicate progression through time. Capital letters label cells. The little barbells indicate outlets between depressions with numbers to indicate their elevations. The order of the outlets on the left represent the outlets' positions (if any) in the priority queue PQ. The tree that is progressively built represents the depression hierarchy. The black lines outlining the white regions indicate elevation, with values along the y-axis. Colors represent labels, and the barbels on the left also indicate the depression number associated with each label color. **(a)** Initialization. This reflects the state at the end of Figure 7. The five outlets have been sorted in order of increasing elevations. Five depressions are in the hierarchy, but none of them are connected yet. **(b)** The lowest outlet (between 3 and 4) is popped. A new meta-depression, labeled 5, is made and becomes the parent of 3 and 4. All cells in 3 and 4 with elevations equal to or greater than the outlet's elevation implicitly become a part of 5. **(c)** The new lowest outlet (between 1 and 2) is popped. A new meta-depression, labeled 6, is made and becomes the parent of 1 and 2. All cells in 1 and 2 become part of 6. **(d)** The lowest outlet is now between 2 and 3. We note that 2 now has a parent and should actually be referred to as 6 (the disjoint-set *DS* accelerates this look-up) while 3 also has a parent and should be referred to as 5. A new meta-depression, labelled 7, is created. **(e)** The lowest outlet is now between 0 and 1. We refer to 1 by its parent's label, 7. 0 is the ocean, so no new meta-depression is made; 7's parent simply becomes 0. **(f)** The outlet between 4 and 0 is the only one left. But 4's parent is already 0, so nothing needs to be done.

tree, thereby ensuring that the height of any tree is logarithmically bounded. While this is computationally advantageous, the downside of "union by rank" is that it relabels the root nodes of trees in a way that would prevent us from building the binary trees of the depression hierarchy. We therefore use disjoint-set without "union by rank".

To determine which set hosts an element, we follow the chain of parents in the disjoint-set from that element upwards until we encounter an element that is its own parent. For the depression hierarchy, this ultimate parent is a cell that contains an *oceanlink*. Critically for computational efficiency, the disjoint set then points all elements to the appropriate root, ensuring that future queries on any element in the path execute in $O(1)$ time, a technique known as "path compression". With the disjoint set in hand, an outlet's depressions can be updated to reflect the current state of the binary tree by querying each depression label in the disjoint set.

We now sort the outlets in order of increasing elevation and loop over them. Let the depressions linked by a given outlet be called $A$ and $B$; $A$ and $B$ are both leaf depressions in the binary tree. Further, let $R(A)$ and $R(B)$ be the result of querying the disjoint-set; that is, $R(A)$ and $R(B)$ are the meta-depressions at the roots of the trees to which $A$ and $B$ belong. Based on this starting point, one of the following three options must be true:

1. $R(A) = R(B)$. In this case, the depressions are already part of the same meta-depression and nothing needs to be done (see Figure 8f).

2. $R(A)$=OCEAN or $R(B)$=OCEAN. Due to the previous condition, only one of these two depressions may link to the ocean.

3. Neither of the above is true. In this case, two depressions are meeting and must be joined into a meta-depression.

For Case 2 above—either $R(A)$=OCEAN or $R(B)$=OCEAN and $R(A) \neq R(B)$—a few additional steps must be taken to properly build the depression hierarchy. First, for simplicity, the algorithm may swap $A$ and $B$ to ensure that $B$ is the depression that links to the ocean ($R(B)$=OCEAN). This means that $R(A)$ will connect to the ocean through $R(B)$. We make a note that $R(A)$ is ocean-linked (linked to the ocean) through $B$, and also geolinked (physically overflows) into $B$. This ensures that flow from $R(A)$ has an opportunity to fill the $R(B)$ tree from the bottom up. In *DS*, $R(A)$ is merged as a child of the ocean. Figure 8d depicts this.

For Case 3 above—$R(A) \neq$OCEAN, $R(B) \neq$OCEAN, and $R(A) \neq R(B)$—the algorithm recognizes that two depressions are meeting and that a meta-depression must be formed. To do so, the algorithm adds a new depression to *DH* with children $R(A)$ and $R(B)$, and performs a similar operation on *DS*. Finally, the algorithm notes that $R(A)$ and $R(B)$ overflow into each other through the current outlet, and that $R(A)$ geolinks to $B$ and $R(B)$ geolinks to $A$. Figure 8b and c depict this.

Once complete, each cell has both a *leaf label* (its original label) and a *top label* (the uppermost label on the depression hierarchy) associated with it. For examples of both labels, see Figures 9 and 10.

## 4   Theoretical Analysis

In computer science, the performance of algorithms can be analyzed based on how they will scale as the amount of data they process increases. In particular, if $f(N)$ is the exact run-time of some complicated algorithm, then $f(N) = O(g(N))$ implies

this run-time has an upper bound of $c \cdot g(N)$ for some constant $c$ and some $N \geq N_0$. The notation $f(N) = \Theta(g(N))$ implies both an upper and lower bound, for appropriate constants. Such bounds are referred to as the *time complexity* or time of the algorithm (Skiena, 2012). This same notation can be used to measure the *space complexity* of an algorithm: the amount of memory it requires.

We apply this to the algorithms described here. Let the number of cells in *DEM* be $N$. The time complexity of finding the ocean is then $O(N)$, since this requires a single pass across the data. Similarly, the time required to find pit cells is $O(N)$. For depression assignment, all $N$ cells must pass through the priority queue. Following Barnes et al. (2014b), we use a radix heap (Akiba, 2015) constructed to have $O(1)$ operations for both integer and floating-point data. Therefore, depression assignment takes $O(N)$ time for both integer and floating-point data. *OC* is a hash table, so additions and accesses are $O(1)$. Additions and accesses to *DS* using only path compression are $\Theta(n + f \cdot (1 + log_{2+f/n} n))$ for $n$ set and $f$ find operations (Cormen et al., pp. 571–572). Since depression merges are always directly preceded by find operations, $n$ and $f$ are small constants, so manipulations on *DS* take $O(N)$ time. Finally, all of the outlets need to be processed in order to build the forest of binary trees. The number of outlets is unknown, but certainly has an $O(N)$ worst case. Therefore, the entire algorithm runs in $O(N)$ space and time.

## 5 An Alternative Design

Using a priority queue, even one that is $O(N)$, serializes the algorithm. Steps 1–8 of the following alternative design can each be parallelized. The design involves three stages: identifying flats, identifying basins, and building the hierarchy. This can be done as follows: (1) Cells are assigned flow directions. (2) Cells without flow directions are identified—these are flats. (3) Each cell in the flat performs a disjoint-set merge with all its neighbours of the same elevation using the cells' array indices as their keys. If a cell's neighbour has a flow direction (meaning that the particular cell is on the edge of the flat), the neighbouring cell is added to a queue and a note is made that this flat can drain. (4) At this point, all flats are represented by the index of a single one of their member cells. If a flat cannot drain, this representative cell is also added to the queue. (5) A breadth-first traversal is begun for the cells in the queue and used to apply shortest-path flow directions to all the flat cells. (6) At this point, all flats either drain to the ocean or a single, unique pit cell. (7) The ocean and each pit cell each have a unique label. A breadth- or depth-first traversal can be used to apply this label to every cell flowing into a given pit cell or the ocean, forming basins. (8) Exactly as above, the lowest outlet between each basin is identified and (9) the depression hierarchy is constructed.

Unfortunately, load balancing the parallel traversals can be non-trivial. Therefore, we include preliminary source code for a parallel implementation here, but defer developing a performant algorithm for future work.

## 6 Applications

Once the hierarchy has been generated, it can be used to rapidly produce a number of outputs of interest. This includes three different methods for DEM preconditioning, such as those used for hydrological calculations: filling depressions, carving

depressions, and depression filtering. In addition, this approach can be used to compute depression statistics and to model water flow across a landscape.

## 6.1   Depression Filling

Depression filling raises the elevation of all cells within a depression to the level of the depression's lowest outlet. This ensures that all cells have a monotonically-descending flow path to the edge of the DEM. Barnes et al. (2014b) review depression-filling algorithms and offer a general algorithm unifying previous work. This has since been accelerated for serial execution (Zhou et al., 2016; Wei et al., 2018) and parallelized for large datasets (Barnes, 2016a).

The depression hierarchy algorithm can be used to perform depression filling by raising each cell $c$ of the DEM to the elevation of its ultimate outlet to the ocean (i.e., the outlets above 11, 12, or 15 in Figure 1, or the elevation of meta-depression 7 in Figure 8). This operation will leave flat areas behind which can be resolved by other algorithms (Barnes et al., 2014a). Alternatively, since the location of the outlet is known, a breadth-first traversal from that point over the depression's cells will yield a drainage surface.

## 6.2   Depression Carving

Depressions can be removed in $O(N)$ time by carving paths from the pit cells of the depression hierarchy's leaves to the ocean. To do so, the elevation of each depression's pit cell should be noted. Since the location of the depression's outlet is known and every cell has been assigned a flow direction, these flow directions can be followed from the outlet to the pit cell. To remove the depression, the flow directions along this path should be reversed (if they flow away from the ocean) or retained (if they flow towards the ocean). Furthermore, once the reversed path has been built, the original DEM can be altered to enforce drainage by traversing the path from the pit cell to the ocean and decrementing each cell along the way, being careful to use a function similar to C++'s `std::nextafter` to prevent floating-point cancellation. This will produce flow fields similar to those resulting from previous works (Braun and Willett, 2013; Lindsay, 2015). See Figure 9 for an example.

## 6.3   Filtering Depressions

Depressions can be selectively removed by traversing the depression hierarchy. Typically, small or shallow depressions are considered to be artifacts; these can be identified by checking whether a depression's area or volume falls below a threshold. If so, the depression can be filled to the level of its outlet or breached (Lindsay, 2015) by using a priority-queue seeded with any of the depression's pit cells in a way that is similar to Priority-Flood (Barnes et al., 2014b). See Figure 9 for an example.

## 6.4   Depression Statistics

The number of cells in a depression, the area the depression covers, and the volume of the depression can all be calculated by adapting the depression-filling method above. To do so, a cell $c$'s elevation is compared with the outlet elevations of the

| Dataset | Dimensions | Cells | Time (s) | Barnes2014 (s) | Zhou2016 (s) | Wei2018 (s) |
|---|---|---|---|---|---|---|
| Madagascar | 2000 x 1000 | $2.0 \cdot 10^6$ | 0.2 | 0.2 | 0.2 | 0.1 |
| U.S. Great Basin | 1920 x 2400 | $4.6 \cdot 10^6$ | 1.0 | 1.0 | 0.9 | 0.4 |
| Australia | 5640 x 4200 | $2.3 \cdot 10^7$ | 2.4 | 2.0 | 2.6 | 1.2 |
| Africa | 9480 x 9000 | $8.5 \cdot 10^7$ | 17.7 | 16.2 | 11.8 | 5.6 |
| N&S America | 18720 x 17400 | $3.2 \cdot 10^8$ | 47.7 | 48.3 | 37.8 | 19.0 |
| Minnesota 30m topobathy | 34742 x 23831 | $8.2 \cdot 10^8$ | 117.3 | 119.7 | 101.4 | 39.4 |
| GEBCO_14 global 30" topobathy | 86400 x 43200 | $3.7 \cdot 10^9$ | 1881.5 | 1879.9 | 1508.5 | 629.1 |

**Table 1.** Datasets used, their dimensions, and algorithm wall-time on the Comet cluster run by XSEDE (see main text for full specifications). Topographic data for Madagascar, the U.S. Great Basin, Australia, Africa, and North & South America, were clipped from the global GEBCO_08 30-arcsecond global combined topographic and bathymetric elevation data set (GEBCO, 2010). The Minnesota 30m topobathy data is the merged result of two data sources. The topography is resampled from the Minnesota Geospatial Information Office's 1m LiDAR Elevation Dataset (Office, 2019). Bathymetric data were provided by the Minnesota Department of Natural Resources (of Natural Resources, 2014). Richard Lively of the Minnesota Geological Survey merged and combined these data sets. The GEBCO_14 global 30" topobathy data set was drawn directly from GEBCO (2014). Wall-times are compared against several depression-filling algorithms, as described in the text.

depressions in the hierarchy. The lowest such depression-containing cell $c$ is identified. This depression's *cell count* is then incremented and the cell's areas and elevation are added to the depression's *summed elevation* and *summed area*.

The foregoing process produces marginal values: the areas, volumes, and cell counts associated uniquely with each node in the depression hierarchy. To generate totals, the values of each depression below a given node in the hierarchy must be summed. To do so, the depression hierarchy is traversed in depth-first fashion from its leaf depressions upwards to the ocean. Each depression's *cell count* $D_c$, *summed elevation* $D_e$, and *summed area* $D_a$ are then the sum of those cells that belong uniquely to the depression (per the above) and those that belong to the depression's children. If the outlet elevation of the depression is $D_o$, the volume of the depression is then given by $D_a(D_c \cdot D_o - D_e)$.

## 6.5 Flow Modeling

When water falls on a landscape, it flows downhill to the pit cells of depressions. Depressions then begin to fill up until they spill over into neighboring depressions. The combined depression then fills until it too spills over. This continues until the water finds an outlet to the sea. The depression hierarchy described here, with its geolinks, has been optimized to model this dynamic process of filling, spilling, and merging.

## 7 Empirical Tests

We have implemented the algorithm described above in `C++17` using the Geospatial Data Abstraction Library (GDAL) (GDAL Development Team, 2016) to read and write data. For efficiency we use a radix heap (Akiba, 2015) and an optimized hash ta-

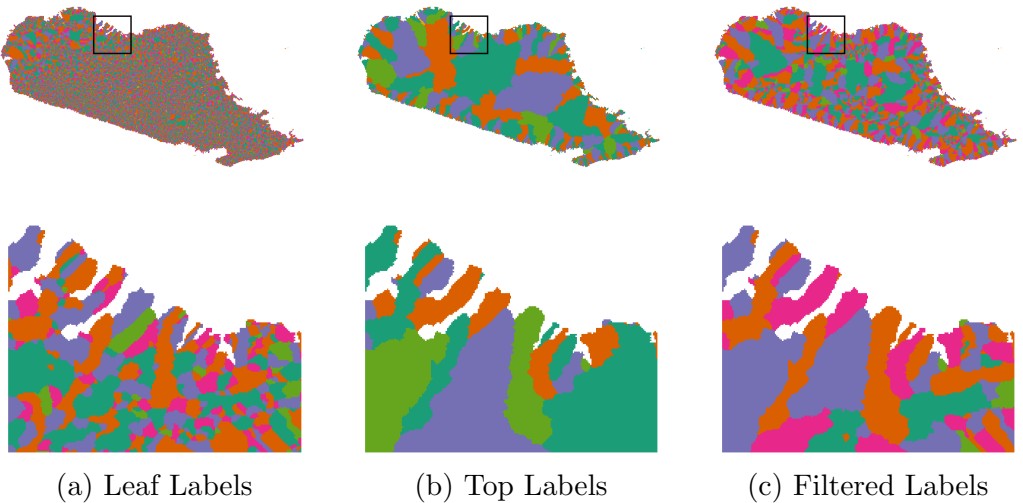

(a) Leaf Labels          (b) Top Labels          (c) Filtered Labels

**Figure 9. Depression hierarchies applied to Madagascar: depression labels.** The *Label* array of the depression hierarchy algorithm is shown here for three situations. The top row depicts all of Madagascar while the bottom row depicts the zoomed areas identified by the black boxes. Since there are too many labels to show in distinct colors the labels have instead been colored so that no two adjacent depressions have the same color using a largest-first greedy algorithm (Kosowski and Manuszewski, 2004; Hagberg et al., 2008). **(a)** depicts the labels assigned to the leaf nodes of the depression hierarchy. **(b)** depicts the labels assigned to the uppermost parent depressions—those which connect directly to the ocean. These are the top-level watersheds of the island. **(c)** depicts the labels after depressions less than a given threshold (30 cells in area) are detected by filtering and removed via carving.

ble (Popovitch, 2019). There are 981 lines of code of which 48% are or contain comments. The code, along with correctness
tests and a makefile, can be acquired from Github (https://github.com/r-barnes/Barnes2019-DepressionHierarchy) or Zenodo (Barnes and Callaghan, 2019).

Tests were run on the Comet machine of the Extreme Science and Engineering Discovery Environment (XSEDE) (Towns et al., 2014). Each node of Comet has 2.5 GHz Intel Xeon E5-2680v3 processors with 24 cores per node and 128 GB of DDR4 RAM. Code was compiled using GNU g++ 7.2.0 with full optimizations enabled. The datasets used and timing results are
shown in Table 1. Datasets were chosen for the large number of depressions they contained. Runtime scales linearly across datasets ranging in size over three orders of magnitude, in agreement with theory. The smaller datasets run quickly enough that they indicate that the depression-hierarchy algorithm may be suitable for use in landscape-evolution models.

Wall-times of the depression hierarchy algorithm are compared against RichDEM's (Barnes, 2016b) implementations of several depression-filling algorithms. The structure of the depression hierarchy algorithm is most directly comparable to the
improved variant of the Priority-Flood algorithm presented by Barnes et al. (2014b) and exhibits almost no overhead in comparison, showing that constructing the depression hierarchy data structure is inexpensive. While running in a comparable amount of time to the Barnes et al. (2014b) algorithm, the depression hierarchy algorithm produces significantly more data on the

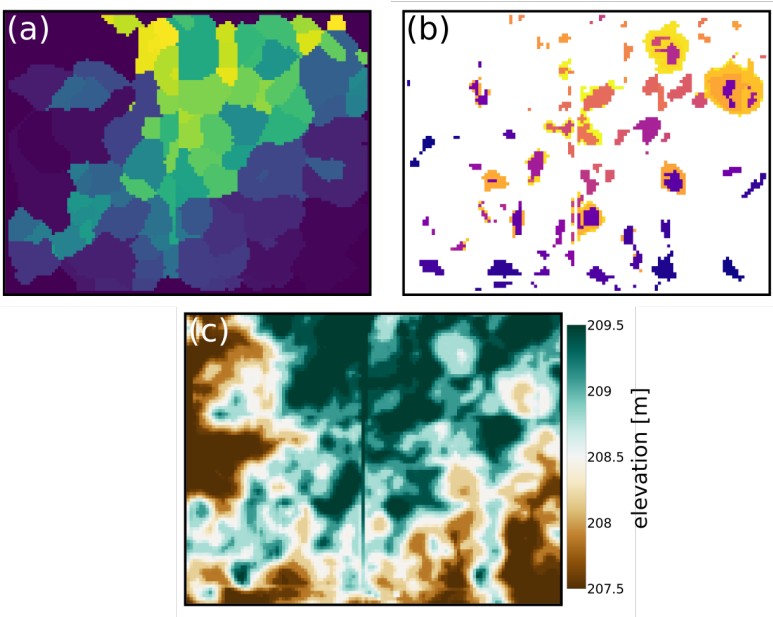

**Figure 10. An example of depression hierarchies** as applied to a small region of the Sangamon River basin, Illinois (see Lai and Anders, 2018). **(a)** The leaf labels assigned to each cell. Each colour (i.e. each leaf label) represents the catchment of a single pit cell within the domain. **(b)** The top labels of cells. Cells that are not part of any depression are coloured white, while coloured cells are part of a depression and have the potential to retain water. **(c)** Land-surface elevation in metres above sea level. Several of the depressions seen in (b) are clearly visible in the topography in (c). The depression hierarchy makes it possible to take these depressions into consideration when considering the hydrology of the region. Traditional depression-filling methods would simply flood all of the depressions without denoting their structural hierarchy or their associated hydrological sub-catchments. The vertical line seen just left of centre is a road, and other roads appear as more subtle vertical and horizontal lines. Panels (a) and (b) show that this road modifies the depression hierarchy, and therefore impacts water and sediment routing.

landscape topology, including individual cell labels and the depression hierarchy data structure itself. Later algorithms from Zhou et al. (2016) and Wei et al. (2018) improve on Priority-Flood by using more complex logic to decrease the number of

cells that need to be processed by the priority queue. Incorporating these improvements into the depression hierarchy algorithm would have made it more difficult to describe and verify, so we do not pursue them here.

## 8    Conclusions

In summary, this paper presents a data structure—the depression hierarchy—that captures the topologic and topographic complexities of depressions in the context of natural landscapes with potential extensions to image processing. The algorithm

used to generate this data structure offers advantages in efficiency, correctness, documentation, and reuseability when com-

pared against previous work. A follow-on paper will describe how the depression hierarchy can be leveraged to accelerate hydrological models and rapidly compute the effects of depression structures on drainage networks.

*Code availability.* Complete, well-commented source code, an associated makefile, and correctness tests are available from Github (https://github.com/r-barnes/Barnes2019-DepressionHierarchy) and Zenodo (Barnes and Callaghan, 2019).

*Author contributions.* KLC and ADW conceived the problem. RB conceived the algorithm and developed initial implementations. KLC and RB debugged and tested the algorithm. RB prepared the manuscript with contributions from all authors.

*Competing interests.* The authors declare that they have no conflict of interest.

*Acknowledgements.* RB was supported by the Department of Energy's Computational Science Graduate Fellowship (Grant No. DE-FG02-97ER25308) and, through the Berkeley Institute for Data Science's PhD Fellowship, by the Gordon and Betty Moore Foundation (Grant
GBMF3834) and by the Alfred P. Sloan Foundation (Grant 2013-10-27).

KLC was supported by the University of Minnesota Department of Earth & Environmental Sciences' Junior F Hayden Fellowship, start-up funds awarded to ADW by the University of Minnesota, and by the National Science Foundation under grant no. EAR-1903606.

Jingtao Lai and Alison Anders provided a copy of their Sangamon River DEM.

Empirical tests and results were performed on XSEDE's Comet supercomputer (Towns et al., 2014), which is supported by the National
Science Foundation (Grant No. ACI-1053575). Portability and debugging tests were performed on the Mesabi machine at the Minnesota Supercomputing Institute (MSI) at the University of Minnesota (http://www.msi.umn.edu).

This collaboration resulted from a serendipitous meeting at the Community Surface Dynamics Modeling System (CSDMS) annual meeting, which RB attended on a CSDMS travel grant.

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
