# Peer review of "Computing water flow through complex landscapes, Part 2: Finding hierarchies in depressions and morphological segmentations"

_Earth Surface Dynamics, 2019_

## Referee Comment (RC1) · Anonymous Referee #1 · 1 Jul 2019

This paper presents an innovative data structure for representing hierarchical depression on complex landscapes and the corresponding algorithm for constructing such data structures. The data structure and algorithm are well presented and explained. It definitely provides insights to hydrological terrain analyses researchers who want to understand and analyze complicated depression groups in a systematically way. However, I think this draft only present the method part of this study without adequate supports from real-world hydrological applications. Although in the Application section, the authors list several potential terrain analyzing processes that this new data structure can be beneficial to, there is no concrete evidence to demonstrate the improvement brought by this new data structure. The only result presented with quantified informa-

tion is Table 1, which only shows the time requirement of implementing this algorithm on data sets in different sizes. To make this paper complete as an individual journal article itself, the authors need to compare the efficiency of running different applications (such as pit filling) w/o introducing this new depression hierarchy structure. Even with another paper submitted, it only focuses on 6.5 Flow Modelling, but evidence for application in section 6.1∼6.4 is still missing. Due to this concern, a major revision decision is recommended to the editors. A set of technical issues and comments for the paper are provided here: 1. If it is possible, try to reconcile the 1-d topographic profiles used in Figure 1&2 and Figure 3&4 as a single dataset/profile. Illustrating the points in the context by jumping back and forth between two examples is confusing. For example, the majority of Section 3.4 Hierarchy Construction is explained with the case presented within Figure 3 and 4. Then in line 12-13 of Page 10, the authors suddenly refer to Figure 1 to illustrate some point. The thing is that the outlet key assignment is only given in Figure 3 and 4. Then the point the authors make ("As an example, in Figure 1, 5 drains into 8, but the cells that actually constitute the outlet will be labeled 2 and 6") is not that obvious to readers. 2. Figure 3(f) "an outlet of elevation 3" A specific elevation number ("3") suddenly appears without any indication in the context. If these numbers need to be maintained, please add a y-axis with labels to the subplot. Also, try to use different number formats (like with circles) to differentiate those representing the PQ popup order from those representing the spilling elevations of the outlets. 3. Page 7 Line 29-30 "Figure 3h-i depicts the front of a traversal, in this case, expanding the area that is defined as OCEAN. We discuss both possibilities below." The placement of this sentence seems odd. It is not closely connected to previous statements in this paragraph, which explains cells assigned with given depression labels. 4. Page 8 Line 23-24 "If any entry for an outlet is already present, only the outlet of lower elevation is retained; this is important, as it allows for the realistic case of multiple spillways that exist between two depressions." This statement seems contradictory. The former part states that the value of the lowest joining cell will overwrite the value in the hash map as the outlet value. Since the value of this hash map is a single value instead of an array. How can it keep track of the multiple-spillway case the authors discuss in the later part? 5. Page 8 Line 24-25 "but the one-dimensional elevation profile in Figure 3 cannot depict the case of multiple outlets of different elevation." Then can you add a figure of a two-dimensional domain to clarify the multiple outlets case? 6. Page 8 Line 28 "assigned each of them a flow direction" As a byproduct, the flow directions are rarely discussed during the depression assignment process, which is understandable. The only place I saw that flow directions were mentioned is in Line 10 (P8): "Flowdir(n) is set to point to c". If I understand it correctly, in this way, the flow directions are assigned locally, which means each cell will drain to the lowest local pit following the assigned directions. This point needs to be emphasized here because they are different from the typical flow directions we have seen draining water to the ocean. 7. Page 9 Figure4(d) "Were M part of another depression (call it 6) that had previously found an outlet to the ocean, then 5's parent would be the depression identified by the label of M, which would be a leaf of the tree rooted by 6. This would ensure that 5 would drain into the bottom of 6 before overflowing out of it." An actual figure could be helpful to illustrate this hypothetical scenario. If the authors think it's not necessary, remove this statement should be fine. 8. Adding a reference to a draft in preparation is not acceptable. Please remove the reference to "Barnes, R., Callaghan, K., and Wickert, A.: Computing water flow through complex landscapes, part 3: Fill-Merge-Spill: Flow routing in depression hierarchies, In preparation, 2019."

---

## Referee Comment (RC2) · Wolfgang Schwanghart (Referee) · 12 Jul 2019

In the second part of their multipart paper, Barnes et al. describe data structures and algorithms to identify and organize internally drained basins in digital elevation models (DEMs). Their approach derives hierarchies of depressions and flow directions. Being a reviewer of the first part of this multipart paper (Callaghan and Wickert, 2019), I commented that water flow through complex terrain with internally drained basins could be tackled using a network of sinks. I am glad to see that the multipart paper develops into this direction, in particular, because the computational advantages are obvious.

Overall, the paper is very well written and organized. It is a very technical paper and

focuses on a thorough description of the developed algorithms. I have no major comments on the algorithms and software implementation. A minor issue is that I found the algorithms easier to understand when reading the captions of the figures. Perhaps, the extensive captions might be better placed in the main text.

My concern is that the paper may be too technical for the readership of ESURF. While I see that the authors are planning a third part that will highlight how the developed software can be used to accelerate hydrological models, I think that the paper would benefit from more illustrations/examples/interpretations of the output of these algorithms. How do sink networks differ between different regions (glacially sculpted low-land regions vs. dryland regions) or different DEMs? Illustrating potential geomorphological applications would be a nice addition to the paper and would considerably widen its readership.

Finally, the empirical tests are done on an high-performance computer. Why? As far as I understand, the code is not (yet) fully optimized for using parallel infrastructure. I wonder how timings of the algorithm would scale on "normal" desktop computer.

All in all, this paper presents an important advance in computational geomorphometry. A demonstration of the geomorphic applications of the developed algorithms would make it even stronger. Such demonstration might be substantial work. Thus, I recommend major revisions although the paper has a very high quality at this stage already.

---

## Author Comment (AC2) · 10 Oct 2019

> A minor issue is that I found the algorithms easier to understand when reading the captions of the figures. Perhaps, the extensive captions might be better placed in the main text.

We have tried to provide multiple ways to understand the algorithms, including the description in the text, pictures via the figures, figure captions, and extensively-commented source code. Our hope is that at least one of these methods will prove effective for each reader. It might be that in your case the figure captions worked best. In the revised paper, we'll try to duplicate or move material from the captions to the

main text where appropriate.

> My concern is that the paper may be too technical for the readership of
> ESURF. While I see that the authors are planning a third part that will
> highlight how the developed soft-ware can be used to accelerate hydro-
> logical models, I think that the paper would benefit from more illustra-
> tions/examples/interpretations of the output of these algorithms. How do
> sink networks differ between different regions (glacially sculpted low-land
> regions vs. dryland regions) or different DEMs? Illustrating potential geo-
> morphological applications would be a nice addition to the paper and would
> considerably widen its readership.

This is a good comment, and indeed one that we wrestled with before deciding to fo-
cus on a more abstract approach. From both this comment and some from Reviewer
1, it seems that at least one specific example would be valuable to demonstrate more
tangibly the application of the depression hierachy to real landscapes. We are con-
sidering two candidates for this example: the Illinois landscape used by Callaghan and
Wickert (2019, a companion paper), and Madagascar, which has diverse topography
but is small enough to allow us to describe its exemplary features without diluting the
technical focus of this paper. Our choice on which to include in the ultimate analysis in
the resubmitted draft will be based on which provides a more useful and intuitive visual
description of the depression hierarchy.

> The empirical tests are done on an high-performance computer. Why? As
> far as I understand, the code is not (yet) fully optimized for using parallel in-
> frastructure. I wonder how timings of the algorithm would scale on "normal"
> desktop computer.

The largest dataset we test requires approximately 15GB of RAM, which is larger than
our laptops (8GB). Since we are located at different institutions, HPC environments

are a convenient way to collaborate. The scaling of the algorithm is unaffected by the compute environment, since this is an instrinsic property of the algorithm.

We thank Dr. Schwanghart for his thoughtful review.

---

## Author Response (AR1)

**Reviewer Response**

**Richard Barnes**

We thank the two referees for their constructive comments on our work. In our response, we have highlighted the reviewer's comments in blue, our response to these comments is highlighted in green, and the changes made to the manuscript are in black.

**1   Reviewer #1**

> I think this draft only presents the method part of this study without adequate support from real-world hydrological applications. Although in the Application section, the authors list several potential terrain analyzing processes that this new data structure can be beneficial to, there is no concrete evidence to demonstrate the improvement brought by this new data structure. The only result presented with quantified information is Table 1, which only shows the time requirement of implementing this algorithm on data sets in different sizes. To make this paper complete as an individual journal article itself, the authors need to compare the efficiency of running different applications(such as pit filling) without introducing this new depression hierarchy structure. Even with another paper submitted, it only focuses on 6.5 Flow Modelling, but evidence for application in section 6.1-6.4 is still missing.

> We will be happy to include some time comparisons for pit filling with and without the depression hierarchy structure in the updated paper (i.e. application 6.1). However, the algorithm does considerably more work than simple pit filling: it produces a data structure that can be used to analyze and operate on nested depressions. Therefore, a direct comparison of the wall-time of the new algorithm versus simple pit filling is not really appropriate: these are separate operations for separate things. This is also true of depression carving.

> We will also include a table with more information about the depression statistics (application 6.4) for the examples processed. These data are retained within the depression hierarchy and would not be available when performing simple pit filling. We will also include an example of depression filtering (6.3) to selectively remove depressions below a certain threshold, and an example of depression carving (6.2).

Table 1 now includes timing comparisons between the depression hierarchy algorithm and a set of depression-filling algorithms. We have added a figure (Figure 6) which depicts the effect of filtering and carving depressions.

> If it is possible, try to reconcile the 1-d topographic profiles used in Figure 1 & 2 and Figure 3 & 4 as a single dataset/profile. Illustrating the points in the context by jumping back and forth between two examples is confusing. For example, the majority of Section 3.4 Hierarchy Construction is explained

with the case presented within Figure 3 and 4. Then in line 12-13 of Page 10, the authors suddenly refer to Figure 1 to illustrate some point. The thing is that the outlet key assignment is only given in Figure 3 and 4. Then the point the authors make ("As an example, in Figure 1, 5 drains into 8, but the cells that actually constitute the outlet will be labeled 2 and 6") is not that obvious to readers.

It was impractical to use the exact same topography (and hence, topology) for all four of these figures, since it was necessary to show several different possible cases in the depression tree in Figure 1. Using this full topography would have made Figures 3 and 4 unwieldy. However, we will experiment with remaking Figures 3 and 4 so that they represent the same topography as seen on the right-hand side of Figures 1 and 2, i.e. the depressions labelled 9–15 in the first two figures. This may make it easier for a reader to follow the changes through these four figures. We will update the references to Figure 1 in these later parts of the text to refer to a similar case in Figure 4, so that the reader does not have to jump back as far. We hope that the point made here will be clearer to a reader when viewing Figure 4, which depicts the colours associated with each depression label.

We have modified Figure 1 for clarity, Figure 2 so that the depressions correspond to those depicted in Figure 1, and Figure 4 so that it shows a worked example of the process being applied to the right-hand side of Figure 2. The text and captions have been updated accordingly. We have also added a new figure (Figure 3) depicting the depressions in a more intuitive form.

Figure 3(f) "an outlet of elevation 3" A specific elevation number ("3") suddenly appears without any indication in the context. If these numbers need to be maintained, please add a y-axis with labels to the subplot. Also, try to use different number formats (like with circles) to differentiate those representing the PQ popup order from those representing the spilling elevations of the outlets.

We have added elevations along the y-axis of the plots in Figures 3 and 4.

We have added elevations along the y-axis of the plots in Figures 4 and 5.

Page 7 Line 29–30 "Figure 3h-i depicts the front of a traversal, in this case, expanding the area that is defined as OCEAN. We discuss both possibilities below." The placement of this sentence seems odd. It is not closely connected to previous statements in this paragraph, which explains cells assigned with given depression labels.

This sentence was referring to a specific case in which cells are assigned the depression label associated with the OCEAN depression. It is one of the possible cases for depression label assignment. Nonetheless, this sentence has now been changed to reflect the changed topography seen in Figure 3.

The sentence has been changed to reflect the new topography generated in response to the reviewers' other comments.

Page 8 Line 23-24 "If any entry for an outlet is already present, only the outlet of lower elevation is retained; this is important, as it allows for the realistic case of multiple spillways that exist between two depressions." This statement seems contradictory. The former part states that the value of the lowest joining cell will overwrite the value in the hash map as the outlet value. Since the value of this hash map is a single value instead of an array. How can it keep track of the multiple-spillway case the authors discuss in the later part?

We have added a sentence to the text clarifying what was meant here. To clarify here, we are referring to cases that would be common in the real world, in which two depressions meet one another at multiple cells. In other words, there is a ridge between two depressions, and each time that a cell along this ridge is processed in the depression hierarchy, it will detect that it is a potential link between the two depressions. Once a potential link has been detected, it will check to see whether an outlet has already been recorded in the hash map. If so, it will replace the recorded value only if the new cell has a lower elevation. In this way, only the true outlet, which has the lowest elevation, is recorded between these two depressions.

We have modified the section to read:

If any entry for an outlet is already present, only the outlet of lower elevation is retained. Two depressions may share a border across multiple cells (i.e. there are multiple potential spillways), but only the location of the lowest outlet is recorded since this is the only location where overflow from one depression to the other would naturally occur.

Page 8 Line 24–25 "but the one-dimensional elevation profile in Figure 3 cannot depict the case of multiple outlets of different elevation." Then can you add a figure of a two-dimensional domain to clarify the multiple outlets case?

We are not sure that a figure is needed for this concept, which is a relatively small part of the overall algorithm, now that it has been further clarified. The multiple outlets case is simply any case in which depression 1 and depression 2 (for example) border one another at more than one single cell, which will often be the case. Any location at which a cell from depression 1 and depression 2 are adjacent to one another is a potential outlet. Each of these potential outlets may have a different elevation. Only the outlet with the lowest elevation is recorded.

No changes were made in response to this comment, as discussed above.

Page 8 Line 28 "assigned each of them a flow direction" As a byproduct, the flow directions are rarely discussed during the depression assignment process, which is understandable. The only place I saw that flow directions were mentioned is in Line 10 (P8): "Flowdir(n) is set to point to c". If I understand it correctly, in this way, the flow directions are assigned locally, which means each cell will drain to the lowest local pit following the assigned directions. This point needs to be emphasized here because they are different from the typical flow directions we have seen draining water to the ocean.

> It is correct that flow directions are assigned such that each cell drains to the lowest local pit following the assigned directions. However, this method of flow direction assignment is not vastly different from other typical flow directions used in other algorithms. While there are some algorithms that always route water to the ocean, for example, those that use a least-cost path to the ocean, 'typical flow direction algorithms simply assign flow direction in the local downslope direction. These algorithms rely on a user having already filled depressions prior to calculating flow directions and performing flow routing. This is the key difference in our method: we are not simply filling all depressions prior to calculating flow across the landscape. Instead, we are particularly interested in what happens within the depressions. In the revised paper we will clarify this point.

The revised paper now states early on, in the definition of variables:

> The algorithm returns flow directions as an output. They are determined in a standard way by requiring that each cell direct its flow in a D8 fashion to the lowest of its eight neighbors. In the case that the lowest neighbor is not unique, one is chosen arbitrarily.

The section the reviewer refers to has been modified to read

> [...] *Flowdir(n)* is set to point to $c$. This ensures that flow follows the path of steepest descent since $c$ is the lowest unexplored cell in the DEM.

> Page 9 Figure 4(d) "Were M part of another depression (call it 6) that had previously found an outlet to the ocean, then 5s parent would be the depression identified by the label of M, which would be a leaf of the tree rooted by 6. This would ensure that 5 would drain into the bottom of 6 before overflowing out of it." An actual figure could be helpful to illustrate this hypothetical scenario. If the authors think its not necessary, remove this statement should be fine.

> This refers to the 'ocean-linked case and is shown in Figure 1, where depression 5 is linked to the ocean via depression 6. However, this caption has now changed due to the changes to Figure 4.

The caption of Figure 4 has been updated and no longer contains the hypothetical the reviewer mentions.

> Adding a reference to a draft in preparation is not acceptable. Please remove the reference to "Barnes, R., Callaghan, K., and Wickert, A.: Computing water flow through complex landscapes, part 3: Fill-Merge-Spill: Flow routing in depression hierarchies, In preparation, 2019."

> We have removed reference to this in-preparation paper. We will restore these references if the in-preparation paper is submitted before we submit our revised draft.
>
> We thank the reviewer for their diligence and detailed comments.

The reference is removed.

**2  Review #2: Dr. Schwanghart**

> A minor issue is that I found the algorithms easier to understand when reading the captions of the figures. Perhaps, the extensive captions might be better placed in the main text.

> We have tried to provide multiple ways to understand the algorithms, including the description in the text, pictures via the figures, figure captions, and extensively-commented source code. Our hope is that at least one of these methods will prove effective for each reader. It might be that in your case the figure captions worked best. In the revised paper, we'll try to duplicate or move material from the captions to the main text where appropriate.

Moving the caption material into the main text seemed to distract from the flow of the explanations there, so we have opted not to change the captions in response to this comment.

> My concern is that the paper may be too technical for the readership of ESURF. While I see that the authors are planning a third part that will highlight how the developed soft-ware can be used to accelerate hydrological models, I think that the paper would benefit from more illustrations/examples/interpretations of the output of these algorithms. How do sink networks differ between different regions (glacially sculpted low-land regions vs. dryland regions) or different DEMs? Illustrating potential geomorphological applications would be a nice addition to the paper and would considerably widen its readership.

> This is a good comment, and indeed one that we wrestled with before deciding to focus on a more abstract approach. From both this comment and some from Reviewer 1, it seems that at least one specific example would be valuable to demonstrate more tangibly the application of the depression hierarchy to real landscapes. We are considering two candidates for this example: the Illinois landscape used by Callaghan and Wickert (2019, a companion paper), and Madagascar, which has diverse topography but is small enough to allow us to describe its exemplary features without diluting the technical focus of this paper. Our choice on which to include in the ultimate analysis in the resubmitted draft will be based on which provides a more useful and intuitive visual description of the depression hierarchy.

We have added a figure (Figure 5) which shows one result of applying the depression hierarchy algorithm to Madagascar.

> The empirical tests are done on an high-performance computer. Why? As far as I understand, the code is not (yet) fully optimized for using parallel infrastructure. I wonder how timings of the algorithm would scale on "normal" desktop computer.

> The largest dataset we test requires approximately 15GB of RAM, which is larger than our laptops (8GB). Since we are located at different institutions, HPC environments are a convenient way to collaborate. The scaling of the algorithm is unaffected by the compute environment, since this is an instrinsic property of the algorithm.

> We thank Dr. Schwanghart for his thoughtful review

As a follow-up on this, in 2015 a near-analogue to the setup used by the Comet computer could be purchased as the HP Z440 Workstation (see https://www.amazon.com/HP-Z440-Workstation-Certified-Refurbished/dp/B07JRDR8QT).

[revised manuscript text omitted]

---

## Author Response (AR2)

**Response to Editor**

Richard Barnes, Kerry Callaghan, Andrew Wickert

We thank the associate editor for their constructive comments, and clarifications to our questions.

**1 Associate Editor**

I agree with both reviewers that the paper is very technical and would benefit from examples. As the authors mention in their answers, they are considering this. Please do. I would also to add a simple flow chart with an explanation of the different steps in the algorithms.

In response to this request, we have added an additional example to the paper (Figure 10). We have also added flowcharts (Figures 4, 5, and 6) showing the steps taken by the algorithm. In addition we have made some minor clarifications and flow improvements in the text.

[revised manuscript text omitted]

---

## Author Response (AR3)

**Reviewer Response**

**Richard Barnes**

We thank for the Editor and Associate Editor for their constructive comments and guidance throughout the submission process. Our response to the latest comments is below.

**1  Editor**

> I am pleased to act on the advice of AE Richard Gloaguen and accept your manuscript for publication in ESurf. I ask you, however, to remedy the artefact in Figure 10, if at all possible.
>
> Your manuscript will now be forwarded to the Copernicus publishers. Your prompt response to their requests for input and verification will help in reducing the time to 'print'.
>
> Thank you for working efficiently with AE Richard Gloaguen and his referees. This interaction has resulted in a good improvement of the manuscript and a very worthwhile product. I look forward to seeing your paper published and hope that you will use the journal again on future occasions.
>
> Best wishes,
>
> Niels Hovius

> The artefact in the DEM is a road (Illinois State Highway 32, USA). We have modified the caption of the figure to clarify this and point out the effect it has on the drainage topography and topology. We think that using the DEM with the road is illustrative of the impacts anthropogenic artefacts can have on drainage basins and the need to consider them in analysis.

[revised manuscript text omitted]